# Magnetic sustentation as an adsorption characterization technique for paramagnetic metal-organic frameworks

Nagore Barroso[1], Jacopo Andreo[1], Garikoitz Beobide[1,2], Oscar Castillo [1,2✉], Antonio Luque[1,2], Sonia Pérez-Yáñez[1,2] & Stefan Wuttke [1,3✉]

Nowadays, there are many reliable characterization techniques for the study of adsorption properties in gas phase. However, the techniques available for the study of adsorption processes in solution, rely on indirect characterization techniques that measure the adsorbate concentration remaining in solution. In this work, we present a sensing method based on the magnetic properties of metal-organic frameworks (MOFs) containing paramagnetic metal centres, which stands out for the rapidity, low cost and in situ direct measurement of the incorporated adsorbate within the porous material. To illustrate this sensing technique, the adsorption in solution of four MOFs have been characterized: MIL-88A(Fe), MOF-74(Cu, Co) and ZIF-67(Co). Our simple and efficient method allows the direct determination of the adsorbed mass, as well as the measurement of adsorption isotherm curves, which we hope will greatly advance the study of adsorption processes in solution, since this method is independent of the chemical nature of the adsorbate that often makes its quantification difficult.

[1] BCMaterials, Basque Center for Materials, Applications and Nanostructures, UPV/EHU Science Park, 48950 Leioa, Spain. [2] Organic and Inorganic Chemistry Department, University of the Basque Country, UPV/EHU, Barrio Sarriena s/n, 48950 Leioa, Spain. [3] IKERBASQUE, Basque Foundation for Science, 48009 Bilbao, Spain. ✉email: oscar.castillo@ehu.eus; stefan.wuttke@bcmaterials.net

Adsorption is widely used for different purposes due to the advantages it offers, such as low-cost and high efficiency. For decades, many efforts have been made to develop suitable adsorbents, but few works have focused on developing suitable direct characterization techniques for adsorption processes in solution that do not rely on specific physicochemical properties of the adsorbate. The greatest advantages of adsorption in the removal and purification of chemicals are the low-cost, high efficiency and ease of operation comparing to conventional methods such as chemical precipitation, ion exchange, liquid-liquid extraction or filtration[1] and the considerable variety of adsorbents available. For this reason, adsorption is a widely used process in different industrial and environmental protection applications such as catalysis, storage and purification of water and air[2]. Therefore, adsorption capacity, kinetics and enthalpy are widely studied in order to analyse and evaluate the performance of the adsorbents[3].

For the study of adsorption properties in gas phase there are reliable characterization techniques, particularly volumetric and gravimetric adsorption experiments[4–6]. Other techniques such as vapour adsorption[7,8], quartz crystal microbalance in gravimetry[9], ellipsometry[10], pycnometry[11] and X-ray reflectometry[12] have been proposed in order to study sorption of gas molecules for thin film characterization[13]. However, if the aim is to study adsorption processes in solution, the available techniques rely on indirect characterization of the adsorption by measuring the adsorbate concentration remaining in solution. In this regard, conventional techniques like ultraviolet spectroscopy[14], chromatography[15] or nuclear magnetic resonance[16], among others, are employed. These techniques provide a direct measurement of the adsorbate molecules remaining in solution, which is normally correlated with the amount captured by the adsorbent, but there are cases where this assumption is not so straightforward. This is the case of many drug molecules, which are aimed to be incorporated in an adsorbent (drug carrier) to develop a drug delivery system[17–20]. The adsorbent would capture targeted molecules but if the drug molecule is not completely soluble, its concentration in solution would remain constant as they are in equilibrium with the solid form of the adsorbate[21,22]. Therefore, a simple and reliable characterization technique focused on the physicochemical properties of the adsorbent, suitable to be used under different chemical conditions, will be a major step towards the development of inexpensive, accurate and reproducible approach for characterizing adsorption processes in solution.

In the vast world of porous materials used as adsorbent[23], an interesting class with intrinsic properties that can be directly correlated with the amount of captured adsorbate are paramagnetic Metal Organic Frameworks (MOFs)[24]. MOFs are a class of three-dimensional (3D) porous crystalline materials built from organic linkers (organic building units, OBUs) and metal centres or clusters (inorganic building units, IBUs) that are connected through strong coordination bonds[25]. Its most outstanding properties include high and permanent porosity, large surface areas (surpassing that of the other adsorbents)[26], structural and functional tenability, high thermal stability, tailorable pores and cavities and high adsorption affinity[27,28]. Furthermore, it is noteworthy that the geometry of the organic unit for a specific IBU will direct the structure and thus determine the topology of the resulting MOF, which plays an important role predicting the topology of MOFs[29]. In this context, reticular chemistry was born, the science that studies the chemistry of linking molecular building blocks through strong bonds to make crystalline frameworks. It offers the possibility to design and synthesize a targeted material with atomic precision becoming MOFs as one of the most attractive materials, as their properties can be easily design and tune[28,30–34].

Although many MOFs contain paramagnetic centres, there are very few characterization techniques that take advantage of the magnetic behaviour of a material to monitor the different physical/chemical parameters. As far as we know, there is only one reported work in which the adsorption of a material is monitored through the magnetic properties of the material. It corresponds to a work by Henkelis et al.[35], in which they show how the lanthanide paramagnetic centre of a MOF transits toward a diamagnetic state upon the adsorption/coordination of $NO_x$ molecules through susceptibility measurements. Despite the very interesting features observed for the interaction between the lanthanide based MOF and $NO_X$ adsorbate molecules, it shows a very narrow application scope, as there are not many molecules that upon coordination are able to perform such kinds of paramagnetic-diamagnetic transformations.

Herein, we present a direct and low-cost characterization technique taking advantage on the intrinsic magnetic properties of paramagnetic MOFs for the in situ study of adsorption processes in solution, which is of broader application. This method does not rely on the complex magnetic interactions that usually only emerge at temperatures well below room temperature. Instead, it is based on the weak attraction exerted by an external magnetic field on paramagnetic materials, as most MOFs are at room temperature. This method was used in a previous work[36], where our group was able to observe the attachment of paramagnetic materials to the pole of an electromagnet as well as the variation of the critical magnetic field. The change depends on the molecular weight of these related compounds, in which the paramagnetic molecular entity remained the same in all cases but the mass of the counterion varied. It was the first time in which such phenomena was described but its potential applications for the quantification of the adsorption capacities of paramagnetic porous materials was not implemented until this work. The simple process (Fig. 1) consist in the suspension of a droplet of dispersed MOF microparticles between the poles of an electromagnet. As the field is decreased, a critical value will be reach, in which the magnetic attraction of the paramagnetic MOF for the magnet is no longer sufficient to maintain the aggregate suspended, making it fall in the test tube. The guest molecules present in the pores of the material varies the critical field value, providing a direct reading of the material adsorption. Four MOFs have been selected containing iron(III), cobalt(II) and copper(II) as paramagnetic metal centres. MIL-88A(Fe) was chosen as an example of a water-stable MOF[37]. It contains $Fe_3O$ trimers connected with fumarate linkers forming a flexible material with interconnected pores and cages of 5–7 Å[38,39]. ZIF-67(Co) is a zeolitic imidazole framework (ZIF) formed by bridging cobalt cations with imidazolate linker anions resulting in a sodalite-type topology (SOD) with pore size of 3.4 Å[40,41]. Finally, two isostructural compounds were chosen, cobalt and copper MOF-74, in order to analyse the effect of the plasticity of the metal centre in their adsorption properties. This MOF contains unsaturated $M^{2+}$ sites located at the edges of 1D hexagonal channel of around 11 Å of diameter[42,43].

## Results and discussions

**Magnetic sustentation experimental setting**. Magnetic sustentation experiments (Fig. 2) are performed using a parallel-aligned dipole electromagnet. The procedure starts with the MOF crystals being added to a test tube full of pure solvent, located between the two poles of the electromagnet at their maximum magnetic field. The particles aggregate and are held suspended at the bottom of the magnet's poles. Once the system is stable, the magnetic field is slowly lowered and the particles start falling down in groups starting from those not directly attached to the glass walls of the

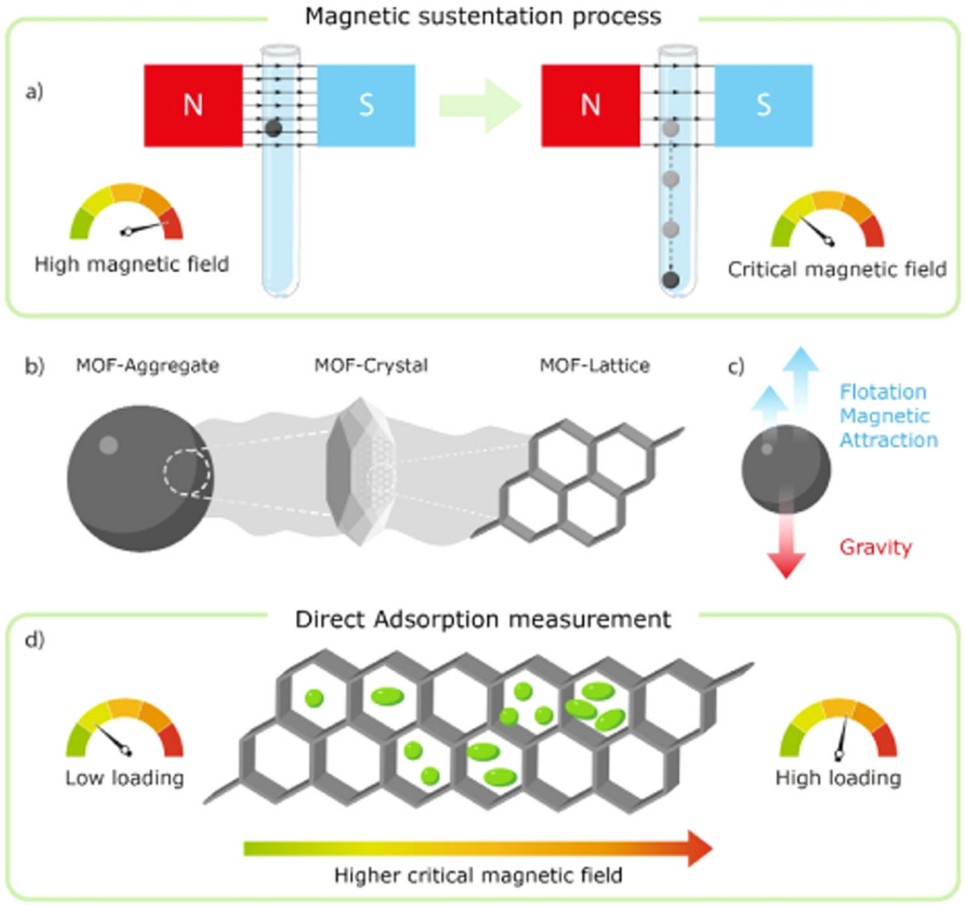

**Fig. 1 Magnetic sustentation process. a** From high magnetic field (suspended aggregate) to the critical magnetic field (aggregate dropping), **b** the MOF aggregate composition, **c** forces acting on the aggregate and **d** effect of the guest loading on the critical magnetic field.

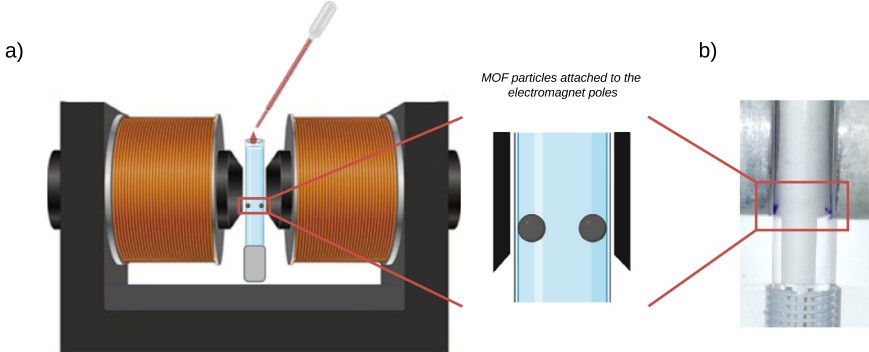

**Fig. 2 Experimental set-up for magnetic sustentation experiment. a** Scheme of the experimental set-up for magnetic sustentation experiment and **b** an example of ZIF-67(Co) particles attached to the bottom part of the electromagnet poles.

test tube (the ones located further away from the magnetic pole). Finally, the last group of particles, those directly attached to the glass wall, fall down. The magnetic field at which this latter detachment happens is what we define as the critical magnetic field. A detailed description of the experimental procedure can be observed in a video provided as supporting information (Supplementary Movie 1). This parameter corresponds to the value of the magnetic field necessary to retain the particles before gravity force overcomes magnetic attraction and the particles are not held suspended to the magnetic poles anymore.

Different solvents were used in these experiments depending on the stability of MOFs; water was used for MIL-88A(Fe), while ethanol was used for MOF-74(Cu), MOF-74(Co) and ZIF-

67(Co). Powder X-Ray diffraction patterns and scanning electron microscopy pictures of the as-synthesized MOFs can be seen in Fig. S1.

**Equations governing the forces balance taking place in the magnetic sustentation experiment**. The magnetic sustentation experiment depicted in Fig. 2, is the result of a fine balance between three forces acting on MOF particles: magnetic attraction, gravitation and flotation. Magnetic attraction is defined as the force that at the bottom of the magnetic pole pushes up the particles toward the maximum magnetic field central area (Eq. 1), gravitation is the force pushing down the particles (Eq. 2) and flotation effect of the solvent is what keeps particles in suspension

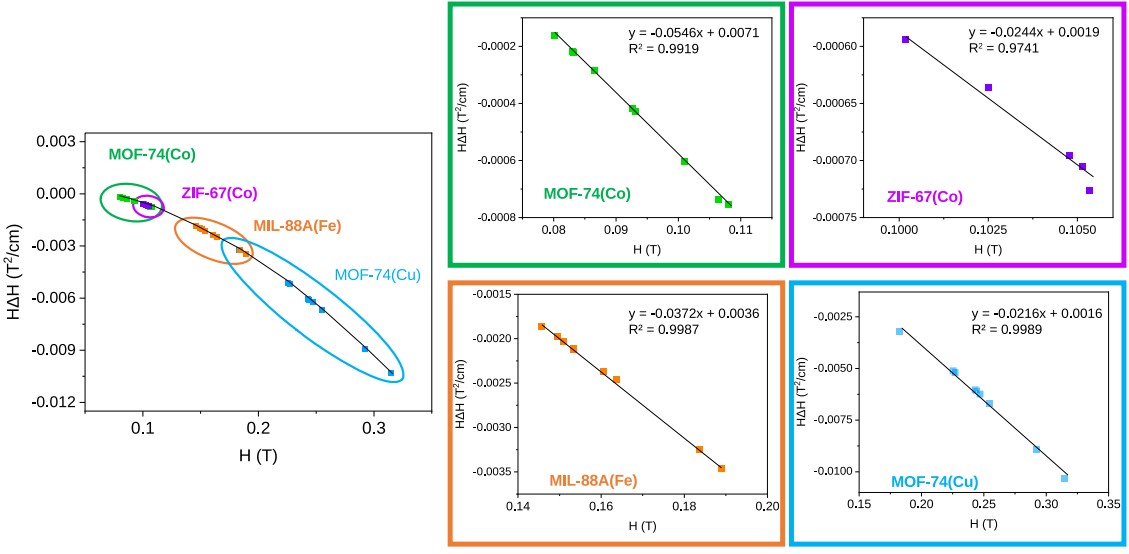

**Fig. 3 Dependence of $H \cdot \nabla H$ vs magnetic field.** $H \cdot \nabla H$ dependence on the magnetic field, $H$, at the centre of the pole and linear fitting in the range at which the magnetic sustentation experiments are performed for each MOF.

pushing up the particles (Eq. 3). The corresponding equations governing each force are depicted below:

$$F_{magnetism} = \nabla(m \cdot H) = m \cdot \nabla H = \frac{\chi_M}{MW_F} \cdot \rho_F \cdot V_F \cdot H \cdot \nabla H \quad (1)$$

$$F_{gravity} = M \cdot g = (M_F + M_M) \cdot g = (V_F \cdot \rho_F + V_M \cdot \rho_M) \cdot g \quad (2)$$

$$F_{flotation} = (V_F \cdot \rho_S + V_M \cdot \rho_S) \cdot g \quad (3)$$

In these equations, we have made an arbitrary separation between the mass, volume and density corresponding to the MOF skeleton ($M_F$, $V_F$ and $\rho_F$) and that corresponding to the adsorbed molecules ($M_M$, $V_M$ and $\rho_M$). The other parameters are described as follows: $\chi_M$ corresponds to the molar susceptibility, H to the magnetic field, $\nabla H$ to the magnetic field gradient and $g$ to the Earth magnetic attraction constant.

The critical magnetic field corresponds to the moment in which the forces pushing the particles upwards and downwards are equal:

$$F_{magnetism} = F_{gravity} - F_{flotation} \quad (4)$$

Replacing Eqs. (1–3) into Eq. (4), Eq. (5) is achieved:

$$\frac{\chi_M}{MW_F} \cdot \rho_F \cdot V_F \cdot H \cdot \nabla H = (V_F \cdot \rho_F + V_M \cdot \rho_M) \cdot g - (V_F \cdot \rho_S + V_M \cdot \rho_S) \cdot g \quad (5)$$

Which can be simplified and rewritten as Eq. (6):

$$\frac{\chi_M}{MW_F} \cdot \rho_F \cdot H \cdot \nabla H = (\rho_F - \rho_S) \cdot g + (\rho_M - \rho_S) \cdot \frac{V_M}{V_F} \cdot g \quad (6)$$

On the other hand, $V_F$ and $V_M$ can be defined as: $V_F = \frac{M_F}{\rho_F} = \frac{n \cdot MW_F}{\rho_F}$ and $V_M = \frac{M_M}{\rho_M} = \frac{n \cdot x \cdot MW_M}{\rho_M}$, in order to provide the relation between the volumes of the adsorbed molecules ($V_M$) and MOF skeleton ($V_F$) as follows:

$$\frac{V_M}{V_F} = \frac{x \cdot MW_M \cdot \rho_F}{MW_F \cdot \rho_M} \quad (7)$$

Replacing Eq. (7) in Eq. (6) we obtain Eq. (8):

$$\frac{\chi_M}{MW_F} \cdot \rho_F \cdot H \cdot \nabla H = (\rho_F - \rho_S) \cdot g + (\rho_M - \rho_S) \cdot \frac{x \cdot MW_M \cdot \rho_F}{MW_F \cdot \rho_M} \cdot g \quad (8)$$

Which can be rewritten as Eq. (9):

$$x \cdot MW_M = \frac{\chi_M \cdot MW_F \cdot \rho_M}{MW_F \cdot (\rho_M - \rho_S) \cdot g} H \cdot \nabla H - \frac{(\rho_F - \rho_S) \cdot MW_F \cdot \rho_M}{(\rho_M - \rho_S) \cdot \rho_F} \quad (9)$$

Taking into account that all parameters except "$H \cdot \nabla H$" and "$x \cdot MW_M$" are constant and assuming that the density, $\rho$, of the different organic molecules is nearly the same, the above equation can be simplified to Eq. (10):

$$x \cdot MW_M = A \cdot H \cdot \nabla H - B \quad (10)$$

We can define as $M_{M(F)} = x \cdot MW_M$, where $M_{M(F)}$ is the mass of adsorbate molecules captured per formula of the framework and obtain Eq. (11), which defines a linear relation between the adsorbed mass and the $H \cdot \nabla H$ parameter at which the particles are detached from the pole of the electromagnet.

$$M_{M(F)} = A \cdot H \cdot \nabla H - B \quad (11)$$

On the other hand, the dependence of the $H \cdot \nabla H$ vs $H$ in the parallel configuration of the electromagnet poles is not linear in the whole range but follows a quasi linear dependence for more limited ranges as those taking place during the magnetic sustentation experiments (Fig. 3). Therefore, the above equation can be rewritten as:

$$M_{M(F)} = A' \cdot H - B' \quad (12)$$

**Experimental verification of the absence of dependence with respect to the particle size.** Magnetic sustentation experiments were performed over MIL-88A(Fe) samples with different particle sizes in order to analyse the influence of different particles sizes (1.5, 4.4 and 5.7 μm) on the critical magnetic field. As shown in Fig. 4, there are no significant differences among the measured critical magnetic field for all samples, concluding there is no

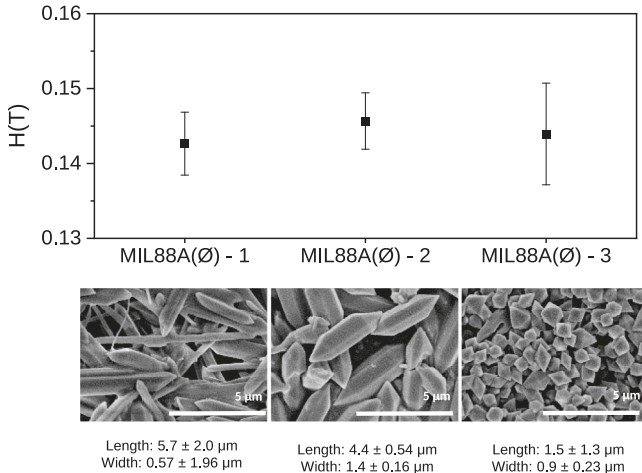

Length: 5.7 ± 2.0 µm
Width: 0.57 ± 1.96 µm

Length: 4.4 ± 0.54 µm
Width: 1.4 ± 0.16 µm

Length: 1.5 ± 1.3 µm
Width: 0.9 ± 0.23 µm

**Fig. 4 Critical magnetic field of three different as-synthesized MIL-88A(Fe) samples.** Determination of the critical magnetic field in water for the three as-synthesized MIL-88A(Fe) samples of different sizes. Each sample was measured five times in order to provide the corresponding associated statistical error.

dependence with respect to the particle size in the range of ~1–6 µm in the considered morphologies. These experimental results agree with theoretical predictions indicating that there should not be such dependence.

**Adsorption selectivity and quantification experiments.** In order to show the suitability of this approach to characterize adsorption processes, different adsorbates were selected, among them, some common organic solvents like DMSO or DMF and drug molecules such as acetylsalicylic acid or sodium naproxen (Fig. 5). Critical magnetic field was determined in the same way as for plain MOFs. Briefly, 15 mg of the MOF were placed in 1.5 mL of the corresponding solvent containing 50 mg of the corresponding adsorbate for 24 h at room temperature and under constant agitation before proceeding with the magnetic sustentation experiment.

In Fig. 6, the measured critical magnetic field after the adsorption experiments can be seen for the selected MOFs (see Tables S1-4 for detailed data). MIL-88A(Fe) and MOF-74(Cu, Co) were able to adsorb some of the tested compounds. More precisely, the smaller the size of the adsorbate, the higher the mass adsorbed and therefore, the magnetic field necessary to retain MOF particles attached to the poles. In that sense, while for smaller organic molecules such as acetonitrile and DMSO the adsorption is significant, bigger drug molecules present a more limited adsorption. On the contrary, for ZIF-67(Co), the adsorption of all adsorbate molecules was negligible due to the small size of the pore windows[40]. In the case of isostructural MOF-74, although a similar trend is shown for both metals in the adsorption of small organic molecules, slight differences are observed in the uptake of drugs. When the metallic centre is copper(II), MOF-74 provides a better adsorption for big size drug molecules like acetylsalicylic acid and 4-aminosalicylic acid, comparing to cobalt(II). This can be attributed to slight differences on the porosity of the two materials. Although cobalt MOF shows a bigger surface area (1327 m²/g and 1126 m²/g for cobalt and copper, respectively), MOF-74(Cu) has a bigger pore volume (0.57 cm³/g for copper vs 0.52 cm³/g for cobalt)[44,45], which in combination with the well-known higher coordination sphere flexibility of copper(II) allows the MOF to accommodate bigger molecules in a more efficiently way. Furthermore, the presence of defective positions in MOFs can also play a significant role on the adsorption selectivity due to the different coordination preferences of Co(II) and Cu(II). However, the observed high adsorption values for some of the adsorbates (in terms of mols per formula of MOF) seem to imply that physisorption is the main driving force in the capture of these adsorbates.

After the adsorption experiments, the stability of the crystalline frameworks of these MOFs was verified with PXRD (Fig. S2) showing no modification with respect to the initial pristine MOFs.

Moreover, it has been experimentally proven that the magnetic field and the mass of captured adsorbate are linearly correlated (Fig. 7), in agreement with Eq. (12). For that purpose, the adsorption experiments were reproduced with MIL-88A(Fe) and MOF-74(Cu, Co) and the adsorption was indirectly estimated from the concentration of the adsorbate remaining in solution using UV-VIS spectroscopy. Specifically, DMSO and DMF were chosen for MIL-88A(Fe) and DMSO and TMU for MOF-74(Cu, Co). The concentrations of these adsorbates were calculated with previously prepared regression lines (Fig. S3, S4). The obtained values represented in Fig. 7 with their corresponding uncertainty, corroborate the theoretically predicted linear correlation and provide a calibration curve that allows to directly quantifying the mass adsorbed of the other molecules that are not quantifiable with UV-VIS from the extrapolation of their critical magnetic field. Furthermore, it can also be observed that the slope of the linear fitting shows also dependence with respect to the transition metal present in the MOF. The higher the number of unpaired electrons, the higher the magnetic susceptibility, the steeper the slope. This becomes evident when comparing MOF-74(Co) and MOF-74(Cu), where the other parameters appearing in Eq. (9) remain unchanged.

**Adsorption isotherm curves.** To illustrate the full opportunities this new technique offers, the adsorption isotherm curves (Fig. 8) were measured using different concentrations (0–40 mg/mL) of the selected adsorbate molecules for the magnetic sustentation experiments. Independently and for comparative purposes, the same adsorption isotherm curves were obtained by UV-VIS spectroscopy (Fig. S5) for quantifiable active adsorbates (DMSO, DMF, TMU). The results showed very similar results from both approaches. MIL-88A(Fe) adsorbs a maximum of 15 and 7% of DMSO and DMF, respectively, by the magnetic sustentation technique and 15 and 8% by UV-VIS spectroscopy. For MOF-74(Cu), the adsorption was calculated to be the same for both techniques, 27 and 14% of DMSO and TMU, respectively. Finally, for MOF-74(Co), adsorption of DMSO and TMU was 21 and 10% determined by magnetic sustentation experiments and 20 and 12% with UV-VIS.

In all cases, it can be seen that adsorption isotherms calculated by magnetic sustentation experiments and ultraviolet spectroscopy are similar, showing this new approach as a versatile technique that allows not only the determination of the adsorbed mass in solution, but also offers the possibility to prepare adsorption isotherms for any molecule. The advantage of the technique is on the one hand, that it offers a fast and direct way of obtaining the data due to the lack of sample preparation and dilutions, and on the other hand, the independence of the chemical nature of the adsorbate. Conventional characterization techniques like UV-VIS spectroscopy are highly dependent on the adsorbate molecules as the adsorbate needs to be active (i.e. contain double bonds) reducing the adsorbates that can be quantified, while this new approach can be directly used with any molecule, independently of its chemico-physical properties. Therefore and in order to prove this, adsorption isotherms for acetonitrile, a non-quantifiable molecule in UV-VIS spectroscopy, were also determined by magnetic sustentation experiment

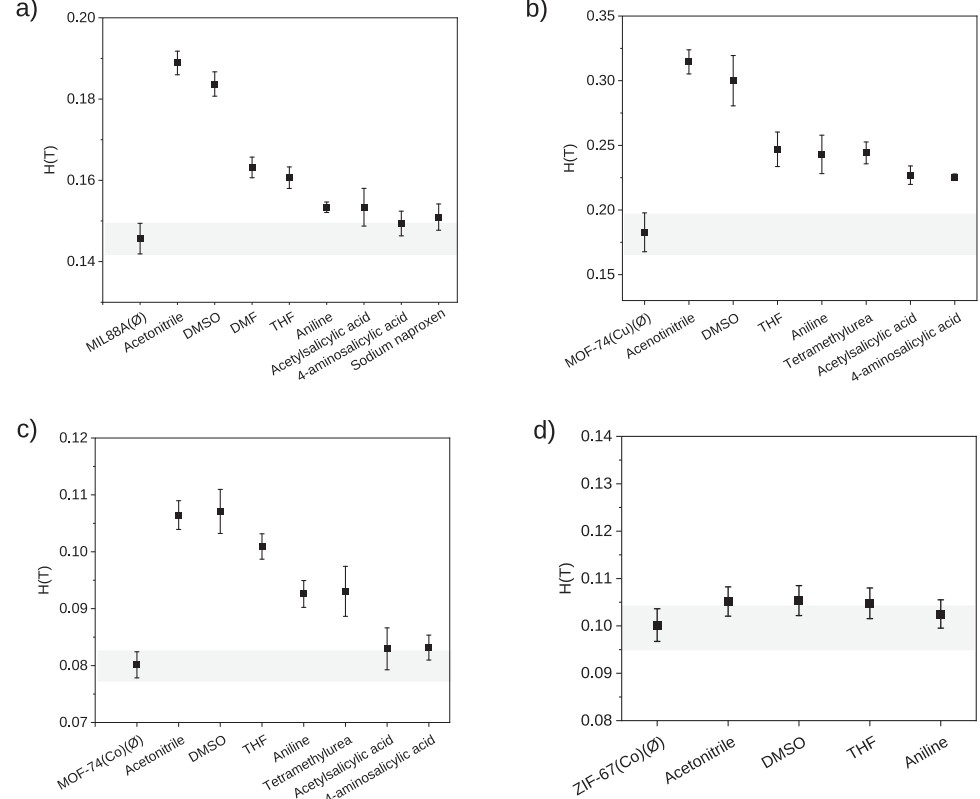

**Fig. 5 Organic compounds.** Selected organic compounds for the adsorption measurements.

**Fig. 6 Critical magnetic field determined by magnetic sustentation experiments.** Determination of the critical magnetic field for: **a** MIL-88A(Fe), **b** MOF-74(Cu), **c** MOF-74(Co) and **d** ZIF-67(Co) after being submerged 24 h and stirred in an aqueous (MIL-88A(Fe)) or ethanol (MOF-74(Cu), MOF-74(Co) and ZIF-67(Co)) solutions containing the adsorbate molecules. Samples labelled with Ø correspond to MOF samples not being exposed to any adsorbate. Each measurement was repeated five times in order to provide the corresponding associated statistical error.

showing a maximum uptake of 17%, 31 and 21% for MIL-88A(Fe), MOF-74(Cu) and MOF-74(Co), respectively.

## Conclusions

In this work, we have presented a fast and direct characterization technique for adsorption processes in solution based on the intrinsic magnetic properties—paramagnetic metal nodes- of MOFs at room temperature. We have developed the theoretical equations that guide magnetic sustentation experiments to better understand the process and experimentally verify its feasibility using four well-known MOFs. On the one hand, water-stable MIL-88A(Fe) was selected, demonstrating that the critical magnetic field

it is not influenced by particles size in the range 1–6 μm. In addition, the adsorption of different organic molecules was quantified, and it was confirmed that there is a linear relationship between the critical magnetic field and captured mass. In order to prove the versatility of this new technique, the same experiments were recorded in ethanol using MOFs that are not water stable, iso-structural Cu(II)- and Co(II)-MOF-74 and ZIF-67(Co). Finally, this technique allowed obtaining adsorption isotherms curves, which altogether provides the advantage of performing a direct and in situ measurement of captured molecules and without the need for any sample preparation that conventional techniques such as ultraviolet spectroscopy usually requires.

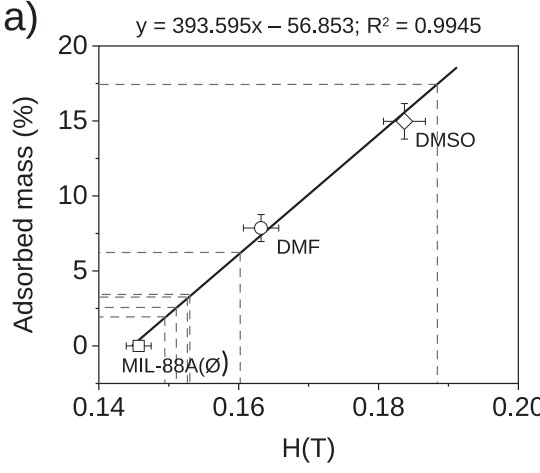

| Calibration Curve - UV-VIS | | H(T) | Adsorbed mass (%) | Adsorbed molecules/ MOF formula |
|---|---|---|---|---|
| ☐ | MIL-88A(∅) | 0.146 | 0 | 0 |
| ◯ | DMF | 0.164 | 7.9 | 0.62 |
| ◇ | DMSO | 0.184 | 15.9 | 1.11 |
| Extrapolated values | | H(T) | Adsorbed mass (%) | Adsorbed molecules/ MOF formula |
| Acetonitrile | | 0.189 | 17.3 | 2.43 |
| THF | | 0.161 | 6.2 | 0.50 |
| Aniline | | 0.153 | 3.3 | 0.20 |
| Acetylsalicylic acid | | 0.153 | 3.3 | 0.11 |
| 4-aminosalicylic acid | | 0.149 | 1.7 | 0.07 |
| Sodium naproxen | | 0.151 | 2.4 | 0.05 |

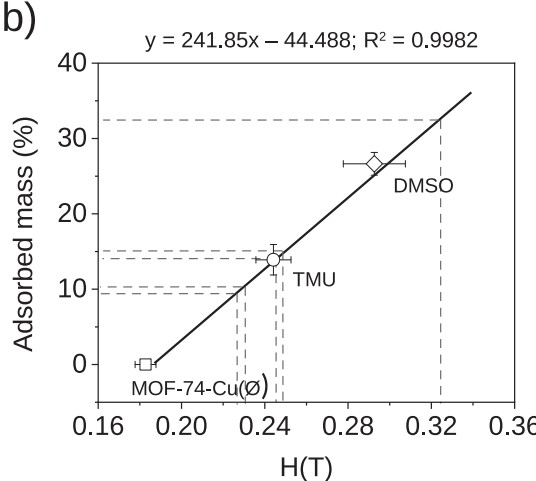

| Calibration Curve - UV-VIS | | H(T) | Adsorbed mass (%) | Adsorbed molecules/ MOF formula |
|---|---|---|---|---|
| ☐ | MOF-74(Cu)(∅) | 0.183 | 0 | 0 |
| ◯ | TMU | 0.244 | 13.9 | 0.39 |
| ◇ | DMSO | 0.293 | 26.3 | 1.10 |
| Extrapolated values | | H(T) | Adsorbed mass (%) | Adsorbed molecules/ MOF formula |
| Acetonitrile | | 0.315 | 31.6 | 2.75 |
| THF | | 0.247 | 13.9 | 0.43 |
| Aniline | | 0.243 | 15.2 | 0.75 |
| Acetylsalicylic acid | | 0.227 | 14.3 | 0.55 |
| 4-aminosalicylic acid | | 0.225 | 10.4 | 0.21 |

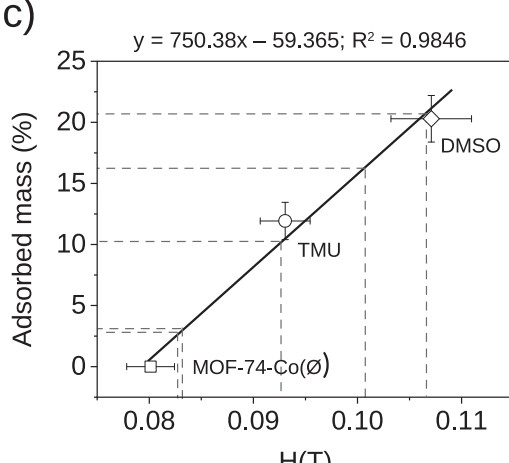

| Calibration Curve - UV-VIS | | H(T) | Adsorbed mass (%) | Adsorbed molecules/ MOF formula |
|---|---|---|---|---|
| ☐ | MOF-74(Co)(∅) | 0.08 | 0 | 0 |
| ◯ | TMU | 0.09 | 11.9 | 0.32 |
| ◇ | DMSO | 0.107 | 20.3 | 0.82 |
| Extrapolated values | | H(T) | Adsorbed mass (%) | Adsorbed molecules/ MOF formula |
| Acetonitrile | | 0.106 | 20.5 | 1.57 |
| THF | | 0.101 | 16.4 | 0.71 |
| Aniline | | 0.093 | 10.1 | 0.34 |
| Acetylsalicylic acid | | 0.083 | 2.9 | 0.05 |
| 4-aminosalicylic acid | | 0.083 | 3.0 | 0.06 |

**Fig. 7 Linear dependence of adsorbed mass (%) vs critical magnetic field.** Calibration curves obtained for quantifiable molecules (DMSO, DMF, TMU) by UV-VIS spectroscopy and the extrapolated values for the other adsorbates represented in a table: **a** MIL-88A(Fe), **b** MOF-74(Cu) and **c** MOF-74(Co). For the calibration curves each measurement was repeated five times in order to provide the corresponding associated statistical error.

## Methods

**Chemicals**. Iron(III) chloride hexahydrate ($FeCl_3.6H_2O$, 99 + % extra pure), fumaric acid (99% +) and aniline (99.5%, extra pure) were purchased from Acros Organics. Cobalt(II) nitrate hexahydrate ($Co(NO_3)_2·6H_2O$, 98%) was purchased from Roth, 2,5-dihydroxiterephthalic acid (DHTPA) and tetramethylurea (TMU) from Fluorochem and dimethylsulfoxide (DMSO, 99%) from Fisher Scientific. Copper(II) acetate hydrate ($Cu(OAc)_2·H_2O$, 98.1%) was purchased from BLDpharm and N,N-dimethylformamide (DMF, 99.9% GLR), methanol (MeOH, 99.8%), acetonitrile (GC HPLC GGR) and tetrahydrofuran (THF, HPLC GGR) from Labkem. Ethanol (EtOH, denatured) was purchased from Chem-labs. Acetylsalicylic acid (ASA), 4-aminosalicylic acid (4-ASA, 99%) and naproxen sodium were purchased from Aldrich. All reagents were used without further purification except for ethanol that was distilled in order to remove the denaturant.

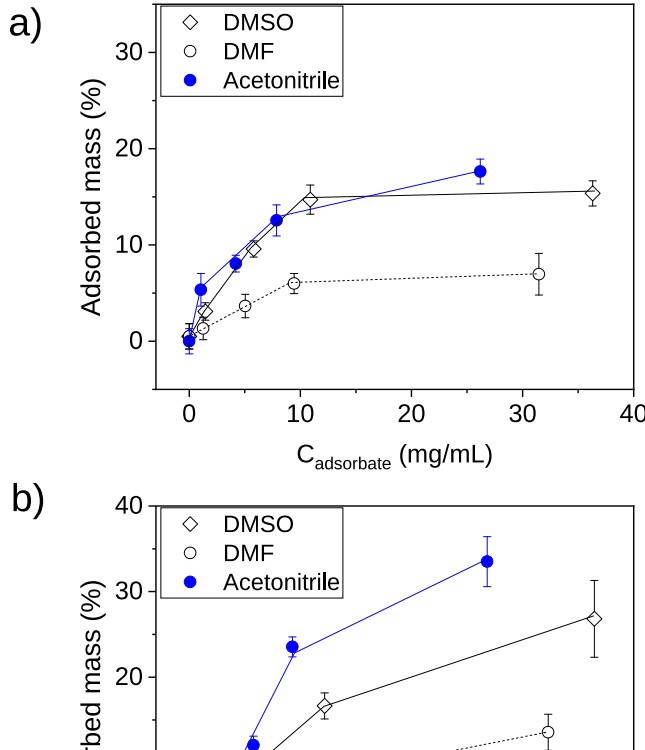

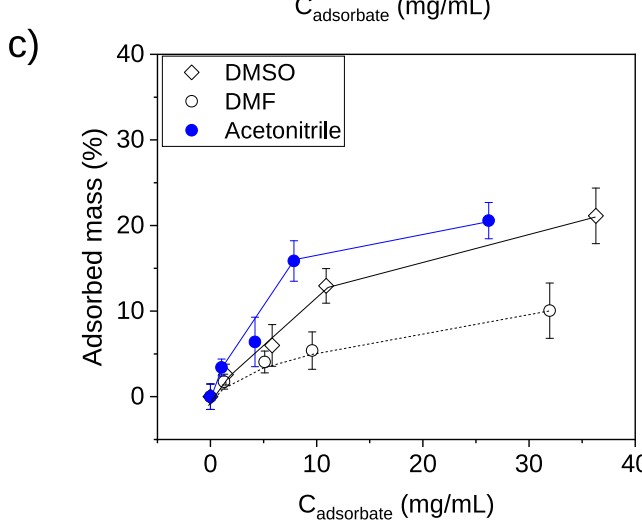

**Fig. 8 Adsorption isotherm curves determined by magnetic sutentation experiments.** Adsorption isotherm curves for: **a** MIL-88A(Fe), **b** MOF-74(Cu) and **c** MOF-74(Co) measured by magnetic sustentation experiments. Each measurement was repeated five times in order to provide the corresponding associated statistical error.

### MOF synthesis

*MIL-88A(Fe).* MIL-88A(Fe) samples with different particle sizes were synthesized. For that, already reported protocols were used with some modifications[39,46]. Briefly, iron(III) chloride hexahydrate (1.381 g, 5.11 mmol) and fumaric acid (0.593 g, 5.11 mmol) were dissolved in a mixture of 25 mL of solvent: MIL-88A-1 in DMF, MIL-88A-2 in DMF:water (1:1, v-v) and MIL-88A-3 in water. The reaction mixture was placed in a preheated oven at 65 °C for a specific time: MIL-88A-1 and

3 for 12 h and MIL-88A-2 for 4 h. The compound was recovered by centrifugation and washed (15,000 rpm, 15 min, 36 mL) x 2 in water and x 3 in ethanol.

*ZIF-67(Co).* ZIF-76(Co) was synthesized according to an already reported procedure without any modifications[47]. Co(OH)$_2$ (0.372 g, 4.00 mmol) and 2-methylimidazole (0.690 g, 8.40 mmol) were hand-grinded thoroughly in order to obtain an homogeneous mixture. Then, the mixture was sealed in a 45 mL Teflon-lined autoclave and heated to 60 °C in 2 h and then to 160 °C in 48 h. The reaction mixture was cool down to room temperature in 12 h. The product was washed with ethanol five times.

*MOF-74(Co).* MOF-74(Co) was synthesized according to an already reported procedure[48]. Cobalt(II) nitrate hexahydrate (0.717 g, 2.41 mmol) was dissolved in a mixture of water, DMF, and ethanol (60 mL, 1:1:1, v-v:v) and then, 2,5-dihydroxyterephthalic acid (0.145 g, 0.73 mmol) was added. The reaction mixture was stirred for a few minutes at room temperature and the autoclave was sealed. The autoclave was put into an oven at 120 °C for 24 h. The solid was recovered by centrifugation and washed (15,000 rpm, 30 min, 6 mL) x 3 in hot DMF. Afterwards, it was washed three times in ethanol and then stored in ethanol over 2 days.

*MOF-74(Cu).* MOF-74(Cu) was synthesized according to an already reported procedure[49]. Copper(II) acetate hydrate (0.204 g, 1.00 mmol) and 2,5-dihydroxyterephthalic acid (0.101 g, 0.51 mmol) were dissolved separately in 7.5 mL of methanol. Copper acetate solution was added dropwise to the linker solution and it was stirred at room temperature for 24 h. The solid was recovered by centrifugation and washed (15,000 rpm, 30 min, 6 mL) x 3 in hot DMF. Afterwards, it was washed three times in ethanol and then stored in ethanol over 2 days.

**Characterization**. The crystallinity and purity of the samples was assessed by powder X-ray diffraction (PXRD). PXRD patterns were collected on a Philips X'PERT powder diffractometer with Cu Kα radiation ($\lambda = 1.5418$ Å) over the $5 < 2\theta° < 40°$ range with a step size of 0.02° and an acquisition time of 2.5 s per step at 25 °C. Scanning electron microscopy (SEM) was used in order to determine particles size and morphology. For that, samples were coated with a thin gold layer and measured on a Hitachi S-4800 scanning electron microscope (150 s, 20 mA, 10 kV, zoom at ×10.000).

Magnetic sustentation experiments were performed using a dipole electromagnet (Newport Pagnell England Electromagnet Type C sourced by a Hewlett Packard 6655 A System DC Power Supply) in order to determine the critical magnetic field. Each measurement was repeated five times in order to provide the corresponding statistical errors.

UV-VIS spectroscopy was used to quantify the concentration of dimethylsulfoxide (DMSO), dimethylformamide (DMF) and tetramethylurea (TMU) remaining in solution. The absorbance was measured with a Cary 60 Agilent spectrometer using a quartz cuvette at 60 nm/min over the range 190 – 250 nm. Regression lines (see Suppplementary information), using the absorbance maximum wavelength, were prepared for dimethylsulfoxide (DMSO, $3.2 \cdot 10^{-4}$–$4.0 \cdot 10^{-3}$ M in water and ethanol), for dimethylformamide (DMF, $1.9 \cdot 10^{-5}$–$2.37 \cdot 10^{-4}$ M in water) and for tetramethylurea (TMU, $3.3 \cdot 10^{-5}$ – $3.4 \cdot 10^{-4}$ M in ethanol).

### Data availability

All relevant data are available from the main authors upon request. There is also available a movie in the Supporting Information detailing the experimental procedure to perform the magnetic sustentation experiment.

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

## Acknowledgements

This work was supported by the Spanish State Research Agency (AEI), Basque Government (KK-2022/00032) and the European Regional Development Fund (ERFD) through the project PID2020–15935RB-C42. Technical and human support provided by SGIker (UPV/EHU, MICINN, GV/EJ, ESF) is also acknowledged.

## Author contributions

N.B. performed the experiments and prepared the manuscript and the supplementary information. O.C., A.L., G.B. and S.P.-Y. designed the experiments and developed the technique. J.A. and O.C. contributed to formal analysis. O.C. and S.W. contributed to funding acquisition, project administration, supervision, and writing—review and editing.

## Competing interests

The authors declare no competing interests.

## Additional information

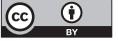

