## [Peer Review File · Communications Chemistry]

Reviewers' comments:

Reviewer #1 (Remarks to the Author):

This paper reports a unique method for getting information on the adsorption of adsorbates supplying from a solution in paramagnetic MOFs. Paramagnetic MOF particles (crystals) are certainly affected by three types of forces in solution, magnetic attraction, gravitation, and floatation. Paramagnetic MOF crystal particles under an attracted magnetic field are able to be retained within sufficient magnetic poles. However, if the particles adsorbed some diamagnetic adsorbates from the solution, the particles will be detached from the pole, because the practical mass of each particle, as well as intrinsic diamagnetic contributions, should be increased, in which the contribution of gravitation becomes much more significant than that of magnetic attraction. Namely, the critical field at which the crystals are detached from the magnetic pole could be related to the adsorbed amounts of adsorbents from a solution medium. Meanwhile, the direct quantification of the adsorbed molecules is indeed much difficult in this technique. Therefore, the calibration curve method using uv spectral change for DMSO/DMF/TMU adsorption was treated for the other larger adsorbates. This technique is unique, but it could be much hard to do actually, because we have to make something of calibration curves based on some of the adsorbates used for the corresponding target paramagnetic MOFs to qualify the adsorption amounts of unknown adsorbates. Anyway, the concept is unique, so this paper may be published in communications chemistry after correcting each point noted below, in particular, the query 7.

- 1) The critical field is determined as a field where the last crystal particle is detached from the pole. However, such visual contacts, i.e., quantification based on the visibility, are quite rough in quantifying techniques. In particular, it could be much hard to do for some cases in small particles. How can you consider their errors in this experiment?
- 2) The data in Figure 4 have some of the error bar. What means of this? Small particles make a large error. How can you determine this error bar experimentally?
- 3) Some of the MOF materials used in this work have ligand-exchangeable sites on the metal ions, e.g., MOF-74. Some adsorbates are able to coordinate to this site, and this coordinating chemistry could be different in cases with different metal ions, although they are isostructural. The argument in Lines 250-255 is wrong, because this effect on guest-coordination was neglected.
- 4) In Line 267-268. What is the mean of linearly in Figure 6. Please give a quantitative line in Figure 6 and explain its mean.
- 5) Line 275-276. The linear correlation and adsorption behavior for the other larger molecules make possible to calculate the molar amounts of adsorbates (mol/mol) in the MOF. All numerical values should be given and discuss about the adsorption properties in pores in the material.
- 6) What is the red plot in Figure 7? The corresponding adsorbate should be indicated.
- 7) For all possible adsorbates were successfully calibrated in Figure 7, right? Therefore, the adsorption isotherms for all adsorbates should be given in Figure 8 (the comparison between two types of techniques should be moved to SI, just justifying the consistency between them). If you had only for DMSO and DMF, this paper should be not worthy to be published, because the quantification for DMSO and DMF adsorbates was indeed achieved indirectly by uv spectral change, not from magnetic field sustentation behavior. This means that this magnetic sustentation method is non-quantifiable; we are able to know only a critical field from this experiment. For qualifying the experimental data, we need a calibration curve such as Figure 7, or another variable parameter such as the mass information of adsorbed adsorbate.

Reviewer #2 (Remarks to the Author):

The manuscript by N. Barroso et al. accounts for a method to measure the mass of an adsorbate

based on the magnetic properties of the MOF. This method was tested with various MOFs (MIL-88A(Fe), MOF-74(Cu,Co) and ZIF-67(Co)) and up to nine different adsorbates (DMF, DMSO, even heavier molecules such as acetylsalicylic acid). The method is based on the linearity between the mass adsorbed and the applied critical magnetic field to keep the MOF sustained in a liquid. The authors also claim the method can be applied to whatever adsorbate, and even use a calibration curve of a magnetic MOF to quantify adsorbates without UV-visible bands.

This method has been previously reported by the authors in reference 35, Cell Rep. Phys. Sci. 2021. Now, in this manuscript the authors extend the method and apply it to the adsorption characterization of MOFs. But there are not so many adsorbates of interest without UV-visible bands and the authors do not test the method with one of them. Thus, a better assessment of the quality of the method for the determination of adsorption isotherms is needed for publication since the methodology was already reported by the authors.

The equations governing the forces balance were better explained in reference 35, so I think the authors should make an effort to write a comprehensible section, with the magnitudes well explained. The units in Figures 7 and 8 are misleading. They should be marked either as % adsorbed or quantity adsorbed in mg/g.

As I understand, the 'calibration curves' obtained in Figure 7 with the aid of the UV spectrometry measurements were used to get the adsorption isotherms of Figure 8. If this is the case, then, UV spectroscopy measurements are always needed to get an adsorption isotherm.

The calibration curves of figure 7 are depicted according to equation 12, but they are better represented like Figures S6 and S7, with the physical magnitude in the y axis and the concentration in the x axis. Usually, the higher the slope of the calibration the better the sensitivity of the method (lower limits of detection), but in Figure 7, MOF-74(Cu) is the MOF with better sensitivity but it has the smallest slope.

What is the limit of detection of the method? what is the smallest quantity adsorbed that produces a change measurable in the critical applied magnetic field? This question arises because the concentration of the analytes in the experiments is quite large (around 50g/L).

An explanation about the changes in the PXRD of the MOFs after adsorption should be added. The MOF material cannot be reused? The MOF is changing its topology? Can they be restored upon removal of the solvent?

Typos, grammatical issues. Line 221: 'Figure'. Lines 256-258: 'copper', the sentence is not well finished.

Reviewer #3 (Remarks to the Author):

The authors present a very thorough accounting in which 4 different MOFs are test subjects to adsorbate solvents in which the magnetic susceptibility changes as it relates to the MOFs and metal centers of the different MOFs. It is well presented and detailed. However, it is not entirely unique, and not well referenced with respect to other work in the use of changes in magnetic susceptibility based on adsorbates for detection. I therefore, unfortunately, have to recommend against publishing this manuscript in Nature Communications Chemistry. I strongly recommend that it be published soon for both the MOF and detection/sensor communities, possibly in ACS Sensors or ACS applied materials and interfaces. More details are listed below.

1) The following recent paper was discovered using google. It also shows the changes in magnetic susceptibility based on adsorbates and whose response was determined by the metal center. Its reference list seems important too. The authors need to reference the work and highlight new discoveries from that already published. See:

ACS Appl Mater Interfaces 2020 Apr 29;12(17):19504-19510.

doi: 10.1021/acsami.0c01813.

2) There is good data collected showing the differences in susceptibility per adsorbate, but very little to no structure-property relationship analysis as to why the change per adsorbate. Why do the authors think this is happening? Is there any predictability for other adsorbates?

3) This is a good candidate for computational modeling to help explain how adsorption, siting, changes to MOF metals, etc are affecting the magnetic response of the adsorbates.

RESPONSE TO THE REVIEWERS

Dear Reviewers,

We sincerely thank you for the thorough analysis of our manuscript and we are glad that the concepts we presented resulted in so many useful and inspiring comments. Please find in the below a point-by-point response to all the comments and a description of the revisions made to the manuscript, which are marked in the manuscript in blue. Several parts of the manuscript text were changed to follow the comments from the Reviewers and Editor, as detailed below.

Reviewer #1 (Remarks to the Author):

This paper reports a unique method for getting information on the adsorption of adsorbates supplying from a solution in paramagnetic MOFs. Paramagnetic MOF particles (crystals) are certainly affected by three types of forces in solution, magnetic attraction, gravitation, and floatation. Paramagnetic MOF crystal particles under an attracted magnetic field are able to be retained within sufficient magnetic poles. However, if the particles adsorbed some diamagnetic adsorbates from the solution, the particles will be detached from the pole, because the practical mass of each particle, as well as intrinsic diamagnetic contributions, should be increased, in which the contribution of gravitation becomes much more significant than that of magnetic attraction. Namely, the critical field at which the crystals are detached from the magnetic pole could be related to the adsorbed amounts of adsorbents from a solution medium. Meanwhile, the direct quantification of the adsorbed molecules is indeed much difficult in this technique. Therefore, the calibration curve method using uv spectral change for DMSO/DMF/TMU adsorption was treated for the other larger adsorbates. This technique is unique, but it could be much hard to do actually, because we have to make something of calibration curves based on some of the adsorbates used for the corresponding target paramagnetic MOFs to qualify the adsorption amounts of unknown adsorbates.

ANSWER: The reviewer has realised the necessity of a previous calibration curve in order to become this technique into a quantitative one. This fact is totally correct as it is correct for many other important analytical techniques; i.e. performing quantification with UV-Vis spectroscopy, chromatography or H-NMR, among others. Therefore, we do not see this as a major drawback of the technique.

Anyway, the concept is unique, so this paper may be published in communications chemistry after correcting each point noted below, in particular, the query 7.

1) The critical field is determined as a field where the last crystal particle is detached from the pole. However, such visual contacts, i.e., quantification based on the visibility, are quite rough in quantifying techniques. In particular, it could be much hard to do for some cases in small particles. How can you consider their errors in this experiment?

ANSWER: Related to the rough nature of the visibility-based quantifying technique, we would like to remember that classical volumetric techniques are based on colour changes produced by the added indicator. Refractometers also rely on a visual reading in order to determine

refractive index or melting point testers for determination of melting point. Therefore, it should not be considered as a drawback of the technique but an improvement opportunity if some kind of image digital processing program could be incorporated into the technique.

Regarding the hardness of determining the point where small particles fall down, we need to emphasise that particles do not drop one by one, but in groups. In such a way that if a big enough quantity of particles has been attached to the pole, those placed further away from the walls of the test tube will fall down first and the last group of particles that will fall are those directly attached to the walls of the test tube. This is usually quite easy to visualize. Perhaps this latter fact can be more easily understood by watching the recorded video that we are providing as SI. In order to provide a better description of the measurements we have rewritten this part of the manuscript: *“The particles aggregate and are held suspended at the bottom of the magnet’s poles. Once the system is stable, the magnetic field is slowly lowered and the particles begin to fall down in groups starting from those not directly attached to the glass walls of the test tube (further away from the magnetic pole) and finally, the last group of particles, those directly attached to the glass wall, fall down. This latter detachment is what determines what we define as the critical magnetic field. A detailed description of the experimental procedure can be observed in a video provided as supporting information (Video S1).”*

In any case and in order to minimize the possible error sources, we need to clarify that a camera with 2.5X magnification is used in the experiments. It is also true that we have found the statistical error being slightly greater in some samples with smaller particle size but this is not attributed to a visualization problem but to adhesion forces taking place between the particles and the glass walls of the test tube, which seems to play a more relevant role for tiniest particles. In any case, we have been able to minimize these effects by agglomerating the particle by centrifugation previous to the measurement.

2) The data in Figure 4 have some of the error bar. What means of this? Small particles make a large error. How can you determine this error bar experimentally?

ANSWER: The assigned errors to each value represented in the corresponding graphs are consequence of the several measurements made for each point (5 times) in order to provide a statistical error. The individual values of those 5 independent measurements are reported in the SI (tables S2-4). We have made clear this part both in the experimental section and in the legends of figures 4, 6 and 8.

The concern about the potential limitation for the measurement of the small particles have been addressed above.

3) Some of the MOF materials used in this work have ligand-exchangeable sites on the metal ions, e.g., MOF-74. Some adsorbates are able to coordinate to this site, and this coordinating chemistry could be different in cases with different metal ions, although they are isostructural. The argument in Lines 250-255 is wrong, because this effect on guest-coordination was neglected.

ANSWER: We agreed with the referee that unsaturated positions in MOFs are usually preferential adsorption sites, especially when performing gas adsorption experiments. In such cases, the nature of the metal centre exerts a strong influence on the adsorption selectivity of these sites. Although, these defective positions are the first ones occupied in MOFs (at least this is what is reported in the literature), the greater amount of captured molecules are due to a physisorption phenomenon, not chemisorption, as it would be the occupation of those uncoordinative positions (having in mind that the experiments are performed in solution, uncoordinative sites are going to be occupied by the solvent molecules and it cannot be ensured that the adsorbate molecules are not going to be able to replace them). Another aspect that points in this direction is the calculation of adsorbed molecules per formula in these experiments being the highest numbers for acetonitrile with values of 1.74 molecules of Acetonitrile per formula of MOF-74-Co and 2.75 molecules of Acetonitrile per formula of MOF-74-Cu, ($M_2(OH)_2(C_8H_4O_6)$) (see inset table in figure 7), that seems to be too high concentration of defects.

We have introduced a sentence in the manuscript explaining this concern between chemisorption and physisorption in these compounds: *“Furthermore, the presence of defective positions in MOFs can also play a significant role on the adsorption selectivity due to the different coordination preferences of Co(II) and Cu(II). However, the observed high adsorption values for some of the adsorbates (in terms of mol per formula of MOF, Fig. S9) seem to imply that physisorption is the main driving force in the capture of these adsorbates”*

4) In Line 267-268. What is the mean of linearly in Figure 6. Please give a quantitative line in Figure 6 and explain its mean.

ANSWER: This graph aiming to provide a qualitative insight of the incorporated adsorbate mass. The greater the deviation of the critical magnetic field with respect to the pristine MOF the greater the mass incorporated. In order to provide quantitative readings a calibration curve is required and that is exactly what we are showing in the updated Figure 7, in which hollow symbols and associated error bars correspond to the samples employed to define the calibration curve, which in fact corresponds to a straight line in accordance with the theoretical predictions described in the manuscript.

5) Line 275-276. The linear correlation and adsorption behavior for the other larger molecules make possible to calculate the molar amounts of adsorbates (mol/mol) in the MOF. All numerical values should be given and discuss about the adsorption properties in pores in the material.

ANSWER: Due to the characteristics of our technique, which depends on the mass of the adsorbate, we can only provide a Figure 7 as adsorbed mass in weight. We have prepared a new version of Figure 7, which shows the calibration curves for each MOF obtained with UV-VIS spectroscopy and the adsorption mass extrapolated data for the UV-VIS not quantifiable molecules, as a table. Here, we indicate the critical magnetic field H(T), the extrapolated mass adsorption percentage and each conversion into mol adsorbate/mol formula of MOF.

6) What is the red plot in Figure 7? The corresponding adsorbate should be indicated.

ANSWER: We have replotted Figure 6 deleting the red plots that were unnecessary to understand the meaning of the Figure. Figure 7 has been replotted as well, as explained above.

a)

Adsorbate molecules	H(T)	Adsorbed mass (%)	Adsorbed molecules/MOF formula
Acetonitrile	0.189	17.3	2.43
THF	0.161	6.2	0.50
Aniline	0.153	3.3	0.20
Acetylsalicylic acid	0.153	3.3	0.11
4-aminosalicylic acid	0.149	1.7	0.07
Sodium naproxen	0.151	2.4	0.05

b)

Adsorbate molecules	H(T)	Adsorbed mass (%)	Adsorbed molecules/MOF formula
Acetonitrile	0.106	31.6	2.75
THF	0.101	13.9	0.43
Aniline	0.093	15.2	0.75
Acetylsalicylic acid	0.083	14.3	0.55
4-aminosalicylic acid	0.083	10.4	0.21

c)

Adsorbate molecules	H(T)	Adsorbed mass (%)	Adsorbed molecules/MOF formula
Acetonitrile	0.315	20.5	1.74
THF	0.247	16.4	0.79
Aniline	0.243	10.1	0.38
Acetylsalicylic acid	0.227	2.9	0.05
4-aminosalicylic acid	0.225	3.0	0.07

Fig. 7 Linear dependence of adsorbed mass (%) vs critical magnetic field. Calibration curves obtained for quantifiable molecules (DMSO, DMF, TMU) by UV-VIS spectroscopy and the extrapolated values for the other adsorbates represented in a table: **a)** MIL-88A(Fe), **b)** MOF-74-Cu and **c)** MOF-74-Co.

7) For all possible adsorbates were successfully calibrated in Figure 7, right?

ANSWER: Only those that were quantifiable by UV-VIS spectroscopy (the pristine MOFs and MOF + DMSO/DMF for MIL88A and MOF + DMSO/TMU for MOF-74(Cu, Co)). The values of the rest of the molecules are obtained from the extrapolation of the measured critical magnetic field using the calibration curves. We have included a table with the extrapolated values, as depicted in the new Figure 7.

Just to avoid any kind of misunderstanding, UV-VIS spectroscopy is employed merely to calibrate those reference materials but it is not fundamental for these new technique as it could be replaced with any other that could provide the same information (chromatography, H-NMR...). Once the calibration curve is determined, the use of the technique is not further necessary.

Therefore, the adsorption isotherms for all adsorbates should be given in Figure 8 (the comparison between two types of techniques should be moved to SI, just justifying the consistency between them). If you had only for DMSO and DMF, this paper should be not worthy to be published, because the quantification for DMSO and DMF adsorbates was indeed achieved indirectly by UV spectral change, not from magnetic field sustentation behavior. This means that this magnetic sustentation method is non-quantifiable; we are able to know only a critical field from this experiment. For qualifying the experimental data, we need a calibration curve such as Figure 7, or another variable parameter such as the mass information of adsorbed adsorbate.

ANSWER: The referee has misunderstood the data from the adsorption isotherm experiments (Figure 8). The adsorption isotherm obtained from magnetic sustentation experiments requires the use of calibration curves appearing in Figure 7, while adsorption isotherm based on UV-VIS spectroscopy were independently performed, just for comparative purposes.

Each adsorption isotherms experiment (magnetic sustentation based ones and UV-VIS based ones) were independently performed using independent calibration curves (that appearing in Figure 7 for the data coming from magnetic sustentation experiments and for UV-VIS the ones plotted in figure S3-4 (previously, Figures S6-7)), which were prepared using solutions of different concentrations to create the corresponding calibration curve. Therefore, the values obtained by each technique cannot be considered as dependent from one another and the superposition of both adsorption curves is a way to ensure the goodness of the new technique in comparison with classical ones. This is, we used UV-VIS quantifiable molecules in order to compare the isotherms with the ones determined from MS.

We must emphasize that in order to obtain an adsorption isotherm is not necessary to perform a double-checking of the data, as explained above. It was simply made for comparative purposes. In order to clarify this fact we are introducing another analyte in

Figure 8 (acetonitrile, which is the molecule with the highest adsorption value for the 3 MOFs), which lacks of a quantifiable signal in UV-VIS spectra.

As requested by the referee, only adsorption isotherms obtained using magnetic sustentation experiment are provided in the manuscript, while those obtained using UV-VIS has been moved to the supporting information (fig S5).

Reviewer #2 (Remarks to the Author):

The manuscript by N. Barroso et al. accounts for a method to measure the mass of an adsorbate based on the magnetic properties of the MOF. This method was tested with various MOFs (MIL-88A(Fe), MOF-74(Cu,Co) and ZIF-67(Co)) and up to nine different adsorbates (DMF, DMSO, even heavier molecules such as acetylsalicylic acid). The method is based on the linearity between the mass adsorbed and the applied critical magnetic field to keep the MOF sustained in a liquid. The authors also claim the method can be applied to whatever adsorbate, and even use a calibration curve of a magnetic MOF to quantify adsorbates without UV-visible bands. This method has been previously reported by the authors in reference 35, Cell Rep. Phys. Sci. 2021. Now, in this manuscript the authors extend the method and apply it to the adsorption characterization of MOFs. But there are not so many adsorbates of interest without UV-visible bands and the authors do not test the method with one of them.

ANSWER: The aim of the paper was to prove the viability of the technique and for that we used well-known paramagnetic MOFs as well as very common organic molecules that can be found in all labs. It is true that many interesting molecules are UV-VIS active, but it is true also that there are a lot of non-active molecules (or at least not so easily quantifiable) by UV-VIS spectroscopy of interest such as alcohols or aminoacids. The aim of our manuscript is to present a new adsorption characterization technique.

Thus, a better assessment of the quality of the method for the determination of adsorption isotherms is needed for publication since the methodology was already reported by the authors.

ANSWER: In ref 36, we reported the first evidence of the dependence of the critical magnetic field required to attach the particles of a paramagnetic material to the pole of the electromagnet with molecular weight in a series of related compounds. In the revised version, we have shown the evolution of this curiosity into a quantification technique for adsorption in solution in porous paramagnetic materials. It represents a relevant step forward in the study of adsorption phenomena in solution for which there is a lack of characterization techniques that provide direct measurements of the adsorbed mass once the material is calibrated. Contrarily to conventional techniques that are based on indirect measurement: the determination of the adsorbate concentration in the remaining solution. We have modified the introductory section to explain better the results appearing in ref. 36 (which was not focused on the quantification of adsorption) and the advances that this new work has provided describing a new characterization technique for the adsorption phenomena in solution.

The equations governing the forces balance were better explained in reference 35, so I think

the authors should make an effort to write a comprehensible section, with the magnitudes well explained.

ANSWER: We have rewritten this part of the manuscript to provide a better understanding of the equations appearing there but we need to emphasize that these equations are not equal to those appearing in ref. 35 (now ref 36).

The units in Figures 7 and 8 are misleading. They should be marked either as % adsorbed or quantity adsorbed in mg/g.

ANSWER: Thank you for this comment and it has been updated, now y axes is called adsorbed mass (%).

As I understand, the 'calibration curves' obtained in Figure 7 with the aid of the UV spectrometry measurements were used to get the adsorption isotherms of Figure 8. If this is the case, then, UV spectroscopy measurements are always needed to get an adsorption isotherm.

ANSWER: The calibration curve for the magnetic sustentation technique was performed using the pristine material and the adsorption values obtained from UV-VIS quantification of the active samples with DMF/TMU and DMSO incorporated. These are the values representing in Figure 7 with hollow symbols and associated error bars. The remaining values were extrapolated from the critical magnetic field using this calibration curve and shown in the table of the same figure.

These calibration curves were also employed to generate the adsorption isotherm curves (in order to calculate adsorbed mass (%)). The adsorption curves were also measured using UV-VIS spectroscopy but only for comparative purpose. In fact, there was no need to for this but taking into account that it is the first time this technique is reported for adsorption, we considered it would be interesting to provide a comparison with a more conventional quantification technique. In addition to that, and to avoid the impression of needing to perform a parallel quantification by means of UV-VIS spectroscopy the adsorption isotherm curves for acetonitrile (a non-active molecule) is also provided. This molecule does not have a quantifiable signal in UV-VIS range.

The calibration curves of figure 7 are depicted according to equation 12, but they are better represented like Figures S6 and S7, with the physical magnitude in the y axis and the concentration in the x axis.

ANSWER: It is usual to represent the known parameter (in this case, the critical magnetic field) in the x axis and the unknown (in this case, the adsorbed mass percentage in this case) in the y axes. This is the reason why we have made these representations. In fact in Figure S3 and S4 that correspond to the calibration curves obtained from UV-VIS, initial concentrations are represented in x axis because it is the known parameter and the absorbance (unknown value) in the y axis.

Eq. 12, which has been theoretically proven says: $M_{M(F)} = A \cdot H - B$

Usually, the higher the slope of the calibration the better the sensitivity of the method (lower limits of detection), but in Figure 7, MOF-74(Cu) is the MOF with better sensitivity but it has the smallest slope.

ANSWER: This is true, but only when you place the unknown (adsorbed mass percentage) in the x axes. In other words, the compounds showing the best selectivity should be that one in which a certain change on the adsorbed mass percentage produce the higher change on the critical magnetic field. In the way the graphs are depicted (percentage of adsorbed mass in y axis and critical magnetic field in the x axis), corresponds to the sample to the smaller slope.

What is the limit of detection of the method? what is the smallest quantity adsorbed that produces a change measurable in the critical applied magnetic field? This question arises because the concentration of the analytes in the experiments is quite large (around 50g/L).

ANSWER: The employed concentration of the analytes in the single adsorption experiments (Figure 6) corresponds to a concentration of the analyte of 50 g/L as explained by the referee. It could seem quite high, but in fact, it only represents a 5 % in mass percentage. In addition to that we have employed far smaller concentrations of the analytes (around 2 g/L) for the preparation of the isotherm curves, for which the technique also works. Finally, concerning the smaller quantity that produce a measurable change it would correspond to double the statistical error of the error (usually, 2-3 %).

The first point of the adsorption isotherms corresponds to the addition of 2 μ L of DMSO/TMU/DMF, so we showed that we are able to detect 1.45 mg/mL, 1.28 mg/mL and 1.26 mg/mL respectively.

An explanation about the changes in the PXRD of the MOFs after adsorption should be added.

ANSWER: Thank you for the comment and we have added a sentence referring to the stability of the crystal structure of the MOFs after the adsorption experiments: *"After the adsorption experiments, the stability of the crystalline frameworks of these MOFs was verified with PXRD (fig. S2) showing no modification with respect to the initial pristine MOFs"*.

The MOF material cannot be reused? The MOF is changing its topology? Can they be restored upon removal of the solvent?

ANSWER: The crystal structure is retained as proved by PXRD, which does not show significant changes, so no topology changes is taking place and the MOF can be remove from the solution, reactivated and reuse again. Regarding to the reusability of the MOF is another question that falls out of the scope of this work. We assume that due to the volatile and small size of some of the molecules employed (acetonitrile, DMSO, DMF and so on) they could be easily removed and let the material ready again for additional adsorption/desorption cycles. However, as states above, in our opinion, it is out of the scope of the work, which is focused on the development of a new characterization technique and not so much on the specific characteristics of MOFs.

Typos, grammatical issues. Line 221: 'Figure'. Lines 256-258: 'copper', the sentence is not well finished.

ANSWER: Corrected.

Reviewer #3 (Remarks to the Author):

The authors present a very thorough accounting in which 4 different MOFs are test subjects to adsorbate solvents in which the magnetic susceptibility changes as it relates to the MOFs and metal centers of the different MOFs. It is well presented and detailed. However, it is not entirely unique, and not well referenced with respect to other work in the use of changes in magnetic susceptibility based on adsorbates for detection. I therefore, unfortunately, have to recommend against publishing this manuscript in Nature Communications Chemistry. I strongly recommend that it be published soon for both the MOF and detection/sensor communities, possibly in ACS Sensors or ACS applied materials and interfaces. More details are listed below.

1) The following recent paper was discovered using google. It also shows the changes in magnetic susceptibility based on adsorbates and whose response was determined by the metal center. Its reference list seems important too. The authors need to reference the work and highlight new discoveries from that already published. See: ACS Appl Mater Interfaces 2020 Apr 29;12(17):19504-19510. doi: 10.1021/acscami.0c01813.

ANSWER: It is an interesting paper where the authors show how a lanthanide paramagnetic centre transits toward a diamagnetic state upon the adsorption/coordination of NO_x molecules. The authors monitor this phenomenon through susceptibility measurements. As stated, it is a very interesting work but with a very narrow application scope as there are not many molecules that upon coordination are able to perform such kinds of transformations. The method reported here is of broad application for paramagnetic porous materials and not limited to NO_x like adsorbates. We consider this paper relevant enough to be introduced in our introductory section because there are few examples of sensing devices based on magnetic properties. In fact, as far as we know it is the only report apart from ours in which magnetic properties are employed for the adsorption characterization. We have introduced a brief discussion of the results reported in this reference and the relation they have with respect to this new develop characterization technique.

2) There is good data collected showing the differences in susceptibility per adsorbate, but very little to no structure-property relationship analysis as to why the change per adsorbate. Why do the authors think this is happening? is there any predictability for other adsorbates?

ANSWER: In our experiments, susceptibility is not directly measured. In fact, molar paramagnetic susceptibility remains constant as far as with normal adsorbate molecules there is no redox processes or high/low spin transitions happening just simple physisorption on the inner surface of the material. The key difference that allows the characterization radicates on the almost equalized magnetic attraction and apparent gravitational force

(gravity – flotation) that allows subtle change of the mass to produce measurable changes on the critical magnetic field at which these two forces exactly equalize.

3) This is a good candidate for computational modeling to help explain how adsorption, siting, changes to MOF metals, etc are affecting the magnetic response of the adsorbates.

ANSWER: Our paper is focused on the in-solution adsorption characterization technique, not on the adsorbing features of the MOFs which have been thoroughly studied in many previous works. Although it would be very interesting, we believe that this kind of work is out of the scope.

REVIEWERS' COMMENTS:

Reviewer #1 (Remarks to the Author):

Almost of the questions arisen for the original manuscript have now been corrected in the revised version. However, I emphasize again here that the qualification of adsorbates is much important in this technique, because of that everybody knows field-dependency of mass and diamagnetic contributions of materials in a filed dipole. In the revised version, figure 7 gave the data for all adsorbates, so all data should be given in the graphs in Figure 7 by extrapolating into the calibration curves with DMF, DMSO, and TMU. These data would have important mean in this work.

Reviewer #2 (Remarks to the Author):

The authors have addressed all the comments raised by the reviewers and the responses are satisfactory.

REVIEWERS' COMMENTS:

Reviewer #1 (Remarks to the Author):

Almost of the questions arisen for the original manuscript have now been corrected in the revised version. However, I emphasize again here that the qualification of adsorbates is much important in this technique, because of that everybody knows field-dependency of mass and diamagnetic contributions of materials in a filed dipole. In the revised version, figure 7 gave the data for all adsorbates, so all data should be given in the graphs in Figure 7 by extrapolating into the calibration curves with DMF, DMSO, and TMU. These data would have important mean in this work.

We have updated Figure 7. For that, we have included the values for the calibration curve obtained from UV-VIS spectroscopy in the table and we have shown in the graph the extrapolated values appearing in the table as requested by the referee.

Reviewer #2 (Remarks to the Author):

The authors have addressed all the comments raised by the reviewers and the responses are satisfactory.

Thank you.